# Unveiling Molecular Mechanisms of Nitric Oxide-Induced Low-Temperature Tolerance in Cucumber by Transcriptome Profiling

**DOI:** 10.3390/ijms23105615

**Published:** 2022-05-17

**Authors:** Pei Wu, Qiusheng Kong, Jirong Bian, Golam Jalal Ahammed, Huimei Cui, Wei Xu, Zhifeng Yang, Jinxia Cui, Huiying Liu

**Affiliations:** 1Key Laboratory of Special Fruits and Vegetables Cultivation Physiology and Germplasm Resources Utilization of Xinjiang Production and Construction Group, College of Agriculture, Shihezi University, Shihezi 832000, China; peiwu0622@163.com (P.W.); chm_agr@shzu.edu.cn (H.C.); xuwei0412@shzu.edu.cn (W.X.); zhifengyangmr@163.com (Z.Y.); 2College of Horticulture and Forestry Sciences, Huazhong Agricultural University, Wuhan 430070, China; qskong@mail.hzau.edu.cn (Q.K.); jrbian2022hzau@163.com (J.B.); 3College of Horticulture and Plant Protection, Henan University of Science and Technology, Luoyang 471023, China; ahammed@haust.edu.cn

**Keywords:** cucumber, low-temperature stress, transcriptome, transcription factor, phenylpropanoid, plant hormones

## Abstract

Cucumber (*Cucumis sativus* L.) is one of the most popular cultivated vegetable crops but it is intrinsically sensitive to cold stress due to its thermophilic nature. To explore the molecular mechanism of plant response to low temperature (LT) and the mitigation effect of exogenous nitric oxide (NO) on LT stress in cucumber, transcriptome changes in cucumber leaves were compared. The results showed that LT stress regulated the transcript level of genes related to the cell cycle, photosynthesis, flavonoid accumulation, lignin synthesis, active gibberellin (GA), phenylalanine metabolism, phytohormone ethylene and salicylic acid (SA) signaling in cucumber seedlings. Exogenous NO improved the LT tolerance of cucumber as reflected by increased maximum photochemical efficiency (Fv/Fm) and decreased chilling damage index (CI), electrolyte leakage and malondialdehyde (MDA) content, and altered transcript levels of genes related to phenylalanine metabolism, lignin synthesis, plant hormone (SA and ethylene) signal transduction, and cell cycle. In addition, we found four differentially expressed transcription factors (MYB63, WRKY21, HD-ZIP, and b-ZIP) and their target genes such as the light-harvesting complex I chlorophyll a/b binding protein 1 gene (*LHCA1*), light-harvesting complex II chlorophyll a/b binding protein 1, 3, and 5 genes (*LHCB1*, *LHCB3*, and *LHCB5*), chalcone synthase gene (*CSH*), ethylene-insensitive protein 3 gene (*EIN3*)*, peroxidase*, phenylalanine ammonia-lyase gene (*PAL*), DNA replication licensing factor gene (*MCM5* and *MCM6*), gibberellin 3 beta-dioxygenase gene (*GA3ox*), and regulatory protein gene (*NPRI*), which are potentially associated with plant responses to NO and LT stress. Notably, HD-ZIP and b-ZIP specifically responded to exogenous NO under LT stress. Taken together, these results demonstrate that cucumber seedlings respond to LT stress and exogenous NO by modulating the transcription of some key transcription factors and their downstream genes, thereby regulating photosynthesis, lignin synthesis, plant hormone signal transduction, phenylalanine metabolism, cell cycle, and GA synthesis. Our study unveiled potential molecular mechanisms of plant response to LT stress and indicated the possibility of NO application in cucumber production under LT stress, particularly in winter and early spring.

## 1. Introduction

Cucumber (*Cucumis sativus* L.) is a typical profitable and nutritious vegetable crop across the world. Despite being a thermophilic species, it is one of the most popular greenhouse-cultivated crops in winter and early spring. However, cucumber plants routinely encounter chilling (0–12.5 °C) during the winter or early spring production in regions with cold climates such as in northern China which appears to be the main limiting factor for the growth, productivity, and quality of greenhouse-grown cucumber [1]. The production of cucumber in greenhouses reaches 17.1 kg·m^−2^, more than five times that of cultivated in the open field, and the productivity of cucumber cultivated in greenhouse decreased by 3–4 kg·m^−2^ due to low temperature (LT) stress [2]. Thus, it is extremely vital to enhance the LT tolerance of cucumber seedlings. According to previous reports, some plant hormones and signaling molecules including auxin (IAA), abscisic acid (ABA), and brassinosteroid (BR) play key roles in improving the LT tolerance of cucumber [3,4]. Nitric oxide (NO), a gaseous signaling molecule, has previously also been shown to be involved in the regulation of plant growth, development, and LT stress response [5,6]. Moreover, it was revealed that exogenous NO increased plant LT tolerance at the physiological level, such as by improving plant photosynthesis, carbohydrate metabolism, antioxidant enzyme activity, and reactive oxygen species (ROS) metabolism [7,8,9,10]. In addition, NO also plays a critical role in plant stress response through mobilizing the flavonoids and other secondary metabolites, such as salicylic acid (SA), ethylene, IAA, ABA, and jasmonic acid (JA), and activating the expression of related genes encoding ABC transporter, glutathione *S*-transferases (GSTs) and cytochromes P450 [11,12,13].

With the development and application of “omics” technologies, transcriptomics analysis has provided a new experimental tool to investigate genome function and its physiological regulation mechanism of stress tolerance in crops. In addition, as a comprehensive and accurate tool, high-throughput RNA sequencing (RNA-seq) provides an opportunity to quantify the transcriptome. Transcriptome profiling based on the RNA-seq is used to analyze gene expression patterns and the molecular mechanism in plant development, response to abiotic stresses and to map the response pathways [14,15,16]. A recent study comprised of both physiological and transcriptome analyses showed that glutathione (GSH) is a downstream signal of hydrogen sulfide (H_2_S)-induced tolerance to chilling stress in cucumber [17]. In a transcriptome analysis of potatoes, the molecular mechanisms of freezing tolerance were found to be modulated by the *ADC1*-associated putrescine pathway, probably by enhancing the expression of the *CBF* gene [18]. More recently, NO was also found to be involved in enhancing melon tolerance to chilling stress by regulating saccharide metabolism, biosynthesis of other secondary metabolites, lipid metabolism, amino acid metabolism, and signal transduction pathways [19]. Those results have greatly extended the current understanding of molecular mechanisms of plant responses to cold stress.

Transcriptional regulation plays a bridging role in plant responses to abiotic stress, and transcription factors (TFs) cover a significant part of the genome and can interact with promoters of different abiotic stress-related genes and initiate the expression of genes [20]. Among the transcription factor signaling cascades, ICE-CBF-COR (COLD-RESPONSIVE) is one of the most well-studied pathways under LT stress. ICEs induce CBF expression through specific binding to MYC cis-acting elements [21,22] and the activated CBF signaling pathways with the induction of *CBF1*, *CBF2,* and *CBF3* expression, followed by the expression of *COR* genes under LT stress [23]. Furthermore, TFs, including members of the AP2 [24], NAC [25], MYB [26], HD-ZIP [27], bHLH [28], CBF [29], and WRKY [16] families were found to participate in plant tolerance against abiotic and biotic stresses by modulating the expression of defense-related genes. For example, transcriptome data in grapevine calli revealed that the expression of genes encoding antioxidant enzymes, such as *peroxidases* and *GST*, was up-regulated after cold treatment in *VaWRKY12*-overexpressing grapevine calli compared to the control calli [30]. In addition, *MYB* was extremely up-regulated in the purple kale and induced by LT stress [26]. Furthermore, HD-ZIP family TFs were also found to be associated with LT stress [31]. However, to our knowledge, few studies have deeply investigated the networks regulated by key TFs in response to LT stress and NO in plants, and the downstream regulation mechanism of these TFs.

Our previous studies at the physiological level have revealed that NO improves the LT tolerance of cucumber by inducing the photosynthetic and antioxidant systems [9,10]. However, the molecular mechanism of cucumber response to LT stress and the NO-induced LT tolerance of cucumbers is still not well understood. In the present study, we aimed to identify the pathways regulated by TFs and their target genes in response to LT stress and NO. Our data suggest that NO plays a significant role in alleviating LT stress by altering the expression of some key TFs that regulate downstream genes involved in LT tolerance. Transcriptome profiling of the current study provides new insights into the NO-regulated LT stress response in cucumbers. The list of abbreviations in the end of the article summarized all of the abbreviations appearing in this study.

## 2. Results

### 2.1. Nitric Oxide (NO)-Induced Low Temperature (LT) Stress Tolerance in Cucumber Seedlings

The severity of cucumber leaf damage was apparently higher in LT treatment than in NO + LT treatment (Figure 1A,B). While LT stress significantly decreased the Fv/Fm in cucumber seedlings compared with the control, NO + LT treatment significantly increased Fv/Fm value in cucumber seedlings compared with the LT treatment (Figure 1D). The image of maximum photochemical efficiency (Fv/Fm) (Figure 1A) was highly consistent with the quantitative value (Figure 1D). Chilling damage index (CI), electrolyte leakage, and malondialdehyde (MDA) content were significantly elevated in cucumber seedlings by LT stress compared with control, while these values were significantly reversed by NO + LT treatment (Figure 1C–F). These results suggested that the inhibitory effect of LT on the growth of cucumber seedlings could be partially reversed by exogenous NO.

### 2.2. Identification of Differentially Expressed Genes (DEGs)

Nine cDNA libraries were established from Jinyan No. 4 cucumber seedlings under normal conditions (Control), low-temperature stress (LT), and a combination of exogenous NO and low-temperature stress (NO + LT) for RNA-seq analysis. The clean reads of each sample ranged from 42.6 M to 55.9 M, with an average amount of 50.5 M. The quality of Q30 bases ranged from 92% to 93%, with an average of 92.9% (Appendix A). In summary, the quality of the data meets the analysis requirements. Salmon was used to align the clean reads of each sample to the annotated gene to estimate the expression abundance (TPM). Furthermore, the mapping ratios ranged from 61% to 71%, with an average amount of 67% (Appendix A). Figure 2A showed the correlation between the samples of the three treatments. Deseq 2 was used to identify DEGs based on the gene expression levels between different samples, and the results are shown in Figure 2B. A total of 7758 DEGs were observed in LT vs. Control, and of these, 4230 were up-regulated and 3528 were down-regulated. In total, 7301 genes were differentially expressed in NO + LT vs. Control, with 3966 up- and 3335 down-regulated. Only 248 genes (151 up- and 97 down-regulated) were differentially expressed in NO + LT vs. LT (Figure 2B). We draw an UpSet diagram for the DEGs obtained from the comparison of the three groups, and the results showed that a total of 1305 and 848 DEGs were uniquely observed in LT vs. Control and NO + LT vs. Control, respectively. Among the 6453 genes differentially expressed in LT vs. Control and NO + LT vs. Control, 121 genes were specifically differentially expressed in NO + LT vs. LT (Figure 2C). These results suggest that the response to LT stress alone and combined treatment of exogenous NO and LT stress may share both the same regulatory pathways and different regulatory pathways in cucumber.

### 2.3. Gene Ontology (GO) Enrichment Analysis of DEGs

In order to understand the function of DEGs in-depth, the DEGs with a corrected *p*-value less than 0.01 (*p*_adj_ < 0.01) were chosen for GO enrichment analysis. The results showed that there were 4609, 4915, and 159 DEGs that obtained GO annotations in the three comparisons of LT vs. Control, NO + LT vs. Control, and NO + LT vs. LT, respectively. In addition, 2439, 2632, and 81 DEGs were enriched in the biological process (BP), 835, 863, and 44 DEGs were enriched in cellular component (CC), and 3320, 3428, and 116 DEGs were enriched in molecular function (MF) in the three comparisons, respectively. Furthermore, the MF contains the most DEGs, followed by BP, and the CC with the least DEGs. Taking *p* < 0.05 as the criterion of significant GO enrichment, the significantly enriched GO terms of each comparison were selected and plotted as a bubble diagram (Figure 3). GO terms associated with DNA-binding transcription factor activity, oxidoreductase activity, transmembrane transporter activity, sequence-specific DNA binding, transferase activity, ADP binding, protein serine/threonine phosphatase activity in MF; transmembrane transport, signal transduction, serine family amino acid metabolic process in BP; and transcription factor complex in CC were significantly enriched in LT vs. Control. Moreover, DEGs including DNA-binding transcription factor activity, transmembrane transporter activity, sequence-specific DNA binding, transferase activity, transferring hexosyl groups, protein serine/threonine kinase activity, protein serine/threonine phosphatase activity in MF; transmembrane transport, serine family amino acid metabolic process, drug transmembrane transport, nucleosome assembly, L-phenylalanine catabolic process, photosynthesis, light harvesting in BP; and transcription factor complex in CC were predominantly enriched in NO + LT vs. Control. In addition, the GO terms including DNA-binding transcription factor activity and oxidoreductase activity in MF; DNA replication initiation in BP; and transcription factor complex, replication fork in CC, were significantly enriched in NO + LT vs. LT.

### 2.4. Kyoto Encyclopedia of Genes and Genomes (KEGG) Enrichment Analysis of DEGs

KEGG enrichment analysis of the DEGs in the three comparisons (LT vs. Control, NO + LT vs. Control, and NO + LT vs. LT) was performed to further elucidate the metabolic pathways and key regulatory genes involved in response to LT stress and exogenous NO. The significantly enriched pathways with *p* < 0.05 were selected for analysis, and the results were shown in Figure 4. A total of 16 pathways were enriched in LT vs. Control, especially in plant hormone signal transduction, phenylpropanoid biosynthesis, MAPK signaling pathway-plant, circadian rhythm-plant, glutathione metabolism, arachidonic acid metabolism, and apoptosis pathways. Fourteen pathways were enriched by analyzing the DEGs in NO + LT vs. Control, in which, phenylalanine metabolism, linoleic acid metabolism, glycolysis/gluconeogenesis, and alpha-Linolenic acid metabolism were significantly enriched. A total of four pathways including phenylpropanoid biosynthesis, phenylalanine metabolism, pentose and glucuronate interconversions, and cell cycle were enriched by analyzing the DEGs between the NO + LT and LT treatments. These results indicated that the signal transduction, carbohydrate metabolism, photosynthesis, circadian rhythm-plant, and glutathione metabolism pathways may be involved in regulating LT tolerance in cucumber seedlings, while NO may improve cucumber LT resistance by activating the differential expression of genes involved in DNA replication and cell cycle pathways.

### 2.5. Prediction and Enrichment of Differentially Expressed Transcription Factors (DETFs) in Cucumber Seedlings

In order to investigate the effect of LT stress and NO + LT treatments on transcription factors (TFs) in cucumber, the TFs encoded by the DEGs and the upstream TFs that regulate the DEGs were mined in this study (Figure 5, Table 1). BLAST was used to align the cucumber protein sequence to the *Arabidopsis* protein sequence for homology comparison after converting the cucumber gene ID to the *Arabidopsis* gene ID. Then the corresponding *Arabidopsis* gene IDs were input into the Plant TFDB (http://planttfdb.gao-lab.org/help_famschema.php, accessed on 20 April 2021) software to predict the TFs. Moreover, The TFs enrichment function was used to perform upstream transcription factors enrichment, and *p* < 0.01 was assigned as a significantly different enrichment. As shown in Figure 5, a total of 504 and 490 DETFs were detected in LT vs. Control and NO + LT vs. Control, and these TFs belong to 53 and 51 families, respectively. The number of DETFs identified in the bHLH family is the largest, followed by ERF, NAC, C2H2, MYB, WRKY, and b-ZIP families in both LT vs. Control and NO + LT vs. Control (Figure 5). Based on the DEGs in LT vs. Control and NO + LT vs. Control, we further identified the upstream TFs that regulated these DEGs. Table 1 showed that 139 and 129 upstream regulatory TFs were enriched in the two comparisons, respectively, of which BBR-BPC, GRAS, AP2, b-ZIP, Dof, BES1, C2H2, MYB, MYB_related, and HD-ZIP were most significant.

### 2.6. Identification of the Key TFs in Response to LT and NO + LT Treatments

Based on the previous enrichment results, the significantly enriched KEGG pathways were screened out, and these enriched DEGs were input into EAT-Up TF v 0.1 (http://chromatindynamics.snu.ac.kr:8080/EatupTF, accessed on 22 April 2021) software to predict the regulatory network. A total of 14 upstream TFs were identified, of which only four TFs including MYB63, WRKY21, b-ZIP, and HD-ZIP were differentially expressed, while b-ZIP and HD-ZIP TFs were specific activation TFs that respond to NO under LT stress. The four TFs potentially regulated 18, 12, 26, and 7 DEGs, respectively (Table 2), and these DEGs were mainly enriched in the flavonoid biosynthesis, phenylalanine metabolism, photosynthesis-antenna proteins, plant hormone signal transduction, cell cycle, and DNA replication (Figure 6B). The DEGs with the same law of change in the up- and downstream of the metabolic pathways were selected to establish the regulatory relationship between the TFs and their downstream target genes (Figure 6A). The genes including light-harvesting complex I chlorophyll a/b binding protein 1 (LHCA1), light-harvesting complex II chlorophyll a/b binding protein 1, 3, 5 (LHCB1, LHCB3, LHCB5), chalcone synthase (CSH) and peroxidase were regulated by MYB63, peroxidase and ethylene-insensitive protein 3 (EIN3) were regulated by WRKY21, gibberellin 3 beta-dioxygenase (GA3oX), DNA replication licensing factor (MCM5, MCM6), phenylalanine ammonia-lyase (PAL) genes were regulated by HD-ZIP, and regulatory protein (NPR1), LHCA1, LHCB1, LHCB3, LHCB5 were regulated by b-ZIP.

### 2.7. Downstream Regulatory Mechanism of Key TFs

To explore the regulatory effects of TFs and their target genes on downstream genes, we further studied the nine target genes of LHCA, LHCB, NPR1, CSH, POX, PAL, MCM5, MCM6, and EIN3, and their upstream and downstream genes in the pathways.

#### 2.7.1. The Regulation of Key TFs on Photosynthetic Antenna Protein-Related Genes

Light-harvesting complex I chlorophyll a/b binding protein (LHCA) and light-harvesting complex II chlorophyll a/b binding protein (LHCB), located at the photosynthetic system I (PSI) and the photosynthetic system II (PSII), respectively, are the most important part of the light-harvesting complex (LHC) in photosynthesis to absorb light energy. As shown in Figure 6A, we identified that the genes LHCA1, LHCB1, LHCB3, and LHCB5 associated with photosynthetic antenna protein were down-regulated in both LT vs. Control and NO + LT vs. Control in this study (Figure 6B). We also found that LHCA1, LHCB1, LHCB3, and LHCB5 were regulated by MYB63 under low-temperature stress, while they were regulated by b-ZIP in NO + LT stress treatment (Figure 6A). In addition, MYB63 was up-regulated, while b-ZIP was down-regulated in both LT vs. Control and NO + LT vs. Control. These results suggested that MYB63 negatively while b-ZIP positively regulated the LHCA1, LHCB1, LHCB3, and LHCB5 to reduce the light energy absorption in cucumber seedlings at LT stress, reduce light damage, and ultimately respond to LT stress.

#### 2.7.2. The Regulation of TF on Genes Related to Flavonoid and Lignin Synthesis

The biosynthesis of flavonoids starts from the phenylalanine metabolic pathway. The cinnamic acid is catalyzed by chalcone synthase (CHS) to form chalcone, and chalcone is catalyzed by chalcone isomerase (CHI) and forms naringenin. Naringenin is catalyzed by flavanone 3-hydroxylase (F3H) to produce flavanone alcohols. Flavonols, the main product of metabolism, enter other different flavonoid synthesis pathways to form different flavonoid substances. In this study, the CHS and CHI were down-regulated in both LT vs. Control and NO + LT vs. Control, while had no significant difference in NO + LT vs. LT (Figure 6B). These results suggested that the synthesis of chalcone and flavanone was reduced by LT, while NO had no effect on the synthesis of chalcone and flavanone under LT stress. The lignin biosynthesis also starts from the phenylalanine metabolism, and the peroxidase, a plant-specific protein, is not only involved in plant hormone metabolism and ROS metabolism but also participates in lignin synthesis in plants. We found that the expression level of peroxidase was up-regulated by LT treatment compared with Control, and exogenous NO further increased the peroxidase expression level under LT stress (Figure 6B). In this study, both CHS and peroxidase were regulated by MYB63, and peroxidase was also regulated by WRKY21. MYB63 and WRKY21 were up-regulated in the three comparisons of LT vs. Control, NO + LT vs. Control, and NO + LT vs. LT. These results indicated that MYB63 negatively regulated the CHS gene, and MYB63 and WRKY21 positively regulated peroxidase at LT stress, leading to decreased synthesis of chalcone and flavanone and increased synthesis of lignin.

#### 2.7.3. The Regulation of TF on Genes Related to Plant Hormone Signal Transduction Pathway

Phenylalanine ammonia-lyase (PAL) can catalyze the formation of trans-cinnamic acid (t-CA), the precursor substance of salicylic acid (SA), from phenylalanine. Exogenous NO could up-regulate the expression of PAL (Figure 6B), which further suggested that NO could promote the accumulation of SA under LT stress in this study. In addition, the expression levels of NPR1 and TGA were up-regulated by LT + NO and LT treatments when compared with the Control. Compared with Control, PR1 was up-regulated by LT + NO, while down-regulated by LT. PR1 was also up-regulated by exogenous NO under LT stress (Figure 6B). Furthermore, HD-ZIP and b-ZIP were down-regulated by LT + NO when compared with Control. PAL and NPR1 were regulated by HD-ZIP and b-ZIP, respectively. These results showed that exogenous NO caused the down-regulation of HD-ZIP and b-ZIP to negatively regulate the expression of PAL and NPR1 genes, thereby promoting SA synthesis, and activating the defense mechanism of the SA pathway.

Ethylene can bind with the ethylene receptor (ETR) located on the endoplasmic reticulum membrane, resulting in the inactivation of the negative regulatory component receptor constitutive triple response 1 (CTR1). The inactivated CTR1 complex can no longer phosphorylate the downstream signal components ethylene-insensitive protein 2 (EIN2), thus causing activation of EIN2. Then the carboxyl end of EIN2 protein (EIN2 CEND) is cleaved followed by entry to the nucleus. EIN2 can promote the accumulation of EIN3/EIL1 in the nucleus by inhibiting the ubiquitination and degradation of transcription factors EIN3-binding F-Box 1_2 (EBF1_2), and then EIN3/EIL1 transcription activates the expression of ethylene-responsive factor1 (ERF1) and other downstream target genes [32,33]. In our study, EIN3, ETR, MKK4_5, and ERF1 were up-regulated by both LT and NO + LT treatments when compared with the Control (Figure 6B), and EIN3 was also regulated by WRKY21 which was up-regulated by LT and NO + LT treatments. These results showed that WRKY21 positively regulated ENI3 to promote EIN3 protein transcription, thereby activating downstream gene expression and ethylene-mediated stress response.

GA_9_ is the initial product of gibberellin, which can be catalyzed to synthesize bioactive gibberellins GA_1_ and GA_4_ by GA3ox. In this study, the GA3ox was down-regulated by exogenous NO under LT stress. In addition, the GA3ox was regulated by HD-ZIP (Figure 6A), while NO down-regulated the expression level of HD-ZIP. These suggested that exogenous NO increased the LT resistance of cucumber seedlings by down-regulating HD-ZIP expression level and further down-regulated the expression level of GA3ox. The reduced active gibberellin delays the growth of cucumber seedlings under LT stress, thereby improving cucumber LT tolerance.

#### 2.7.4. The Regulation of TFs on Genes Related to Cell Cycle Pathway

The MCM node is the DNA replication permission factor, and the MCM2-7 complex is the necessary replication helicase for the initiation and extension of DNA replication in eukaryotic cells. Compared with Control, MCM5 and MCM6 were down-regulated by LT treatment. Furthermore, compared with LT stress, the expression levels of MCM5, and MCM6 were up-regulated, while HD-ZIP, a TF regulating the MCM5, MCM6 was down-regulated by NO + LT treatment (Table 2, Figure 6A). The results showed that exogenous NO increased the LT tolerance in cucumber seedlings by down-regulating HD-ZIP, and HD-ZIP further negatively regulated the MCM5 and MCM6 to promote cell mitosis and increased the cell cycle transition.

### 2.8. Transcript Level Analysis of NO-Induced LT Response Genes

To further validate our transcriptome results, transcripts of 12 key genes involved in plant hormone signal transduction, phenylalanine metabolism, and cell cycle were analyzed by qRT-PCR assays. The expression levels of these 12 genes under each treatment were calculated based on the 2^−^^△△CT^. Comparisons of the relative expression levels of these genes evaluated by the Log_2_ (Fold Change) and qPCR methods are shown in Figure 7. The qRT-PCR results were highly consistent with the transcriptome data, suggesting high confidence in the RNA-seq data and supporting the mechanisms of NO-induced chilling tolerance in cucumber seedlings.

## 3. Discussion

Low temperature (LT), a major environmental constraint, severely affects plant growth and development. It also can lead to a substantial reduction in production, and even plant death [34]. To cope with LT stress, plants have evolved a series of mechanisms at both the physiological and molecular levels. Over the past two decades, numerous components, including messenger molecules, such as NO, calcium (Ca^2+^), hydrogen sulfide (H_2_S), protein kinases, phosphatases, and transcription factors had been identified in cold-stress signaling pathways [9,10,34,35,36]. Our study also showed that LT caused a certain degree of damage to cucumber seedlings, and NO can effectively alleviate this damage (Figure 1). In addition, we employed transcription factors analyses based on transcriptomics to understand the modulation in gene induction and transcription by exogenous NO in LT tolerance in this study.

The RNA-seq technology has the advantage of global representation and precise measurement of the expression level of each gene in a sample by mapping short DNA sequences on a reference [17]. Previous studies intended to identify underlying molecular genetics pathways imparting LT stress tolerance using RNA-seq focusing on the seedling stage of various plants [17,19]. In this study, a total of 7758, 248, and 7301 DEGs were found in LT vs. Control, NO + LT vs. LT, and NO + LT vs. Control, respectively (Figure 2). Previous findings on LT tolerance and NO-induced LT tolerance also revealed a relatively large number of DEGs in melon [19]. Furthermore, a comparative analysis of DEGs of NO + LT vs. LT and common (the genes that differentially expressed in both LT vs. Control and NO + LT vs. Control.) revealed 121 DEGs were exclusively present in NO + LT vs. LT and 127 DEGs were commonly involved in both NO + LT vs. LT and common (Figure 2). This suggested that the presence of these genes may confer tolerance to cucumber during exposure to LT stress. Similar results were also found in NO-induced LT tolerance in melon [19]. These results suggested that the plants may have both the same regulatory and different regulatory pathways when responding to LT stress alone and combined treatment with exogenous NO and LT stress.

The GO and KEGG enrichments are used to analyze the function of genes and the metabolic pathways of plants in response to abiotic stress [13,37,38]. In this study, the GO and KEGG enrichment analyses were also performed to further study the function of these DEGs and analyze the different regulatory pathways when cucumber seedlings were subjected to LT stress and the NO-induced response mechanism in cucumber seedlings. The GO analyses were performed to categorize the genes into three categories, such as molecular function, biological process, and cellular component. The previous studies have shown that the main functions of DEGs in response to LT stress were cell part, organelle, DNA binding, transporter activity, transferase activity, metabolic process, cellular process, single-organism process, cellular protein modification process, and so on [39,40]. In our study, we found that the DEGs responsive to LT stress were mainly enriched in transmembrane transport, signal transduction, and serine family amino acid metabolic process in BP, transcription factor complex, protein serine/threonine phosphatase complex in CC, and DNA-binding transcription factor activity, transmembrane transporter activity, sequence-specific DNA binding in MF according to GO enrichment analysis. The difference in DEGs functions participating in LT stress might be related to different plant species and organs. Consistent with the previous study [19], the DEGs that respond to NO under LT stress were mainly enriched in DNA-binding transcription factor activity, DNA replication initiation, and transcription factor complex. However, the DEGs in NO + LT vs. Control were specifically enriched in transferase activity, transferring hexosyl groups and photosynthesis, and light-harvesting. These results suggested that these DEGs responded to both LT stress and NO. In KEGG enrichment analysis, except for the phenylpropanoid biosynthesis, and plant hormone signal transduction pathways enriched in the previous studies [19,38], the arachidonic acid metabolism, glutathione biosynthesis, circadian rhythm-plant were also significantly enriched in LT vs. Control. Notably, the linoleic acid metabolism and glycolysis/gluconeogenesis pathways were specifically enriched in NO + LT vs. Control. In addition, the pentose and glucuronate interconversions and cell cycle pathways were specifically enriched in NO + LT vs. LT. Therefore, we propose that NO-induced LT tolerance in cucumber seedlings is associated with the regulation of the cell cycle, secondary product metabolism, and plant hormone signal transduction pathways in our experimental conditions. In order to confirm our hypothesis, we analyzed the TFs that respond to NO under LT stress.

As a core of transcriptional regulatory networks, TFs are responsible for the control of gene transcription, technically, TFs also act as the on/off switch of gene expression and are responsible for inducing and repressing genes by binding directly to the promoters region of the target genes in a sequence-specific mode, as well as responding to signal transduction, thereby regulating their function under abiotic stress [41,42]. Thus, identification and evaluation of TF genes related to stress tolerance are essential for understanding the mechanism of cold tolerance in cucumbers. Previous studies have reported that WRKY TFs are involved in cold tolerance in many plants, such as banana fruits, *Coffea canephora*, *Solanum lycopersicum*, *Camellia sinensis,* and tall fescue [43,44,45,46,47]. In addition, MYB, b-ZIP, Dof, NAC, and HD-ZIP were also reported in plant response to LT, such as the *BoPAP1*, an orthologous gene of *BoMYB1*, was induced by LT and subsequently activated a subset of anthocyanin structural genes [26], NAC, MYB, b-ZIP have been reported in response to LT stress in tall fescue [45], HD-ZIP has been shown playing an important role in mediating the resistance to various abiotic stresses [27], and *Dof* is up-regulated in response to cold stress in grapevine [48]. In this study, we focus on the regulatory role of TFs in plant defense and how NO plays a role in translating its bioactivity to recruit these TFs. Consistent with the previous study, we found that a majority of the TF genes were expressed in cucumber leaves under LT stress. We identified 504 and 490 TF genes belonging to 53 and 51 families in LT vs. Control and NO + LT vs. Control, respectively, in which the BBR-BPC, GRAS, WRKY, AP2, b-ZIP, Dof, BES1, C2H2, MYB, MYB-related, and HD-ZIP were significantly enriched. Furthermore, the MYB63, WRKY21, b-ZIP, and HD-ZIP were found to significantly regulate several pathways, such as phenylalanine metabolism, lignin, cell cycle, antenna proteins, and plant hormone signal transduction. Based on these results, we further studied the regulation network of TFs on these metabolic pathways.

Mutations that inhibit phenylalanine ammonia-lyase (PAL) synthetase are usually associated with significant changes in the levels of many phenylpropanoids [49]. As a primary metabolite, phenylalanine is derived from the deamination of phenylalanine precursor through PAL, resulting in cinnamate biosynthesis [50], and cinnamate is a substrate for many different secondary metabolites, including phenylpropanoids, flavonoids, and the cell wall lignin, in which lignin is one of the most important polymers of the plant cell, as well as a wide range of phenolic secondary metabolites. The final stage of lignin biosynthesis is the polymerization of monolignols, which occurs with the participation of peroxidases [51]. Studies of genes encoding peroxidase in *Arabidopsis* have demonstrated a close relationship between these enzymes and lignin accumulation in secondary cell walls [52]. In this study, the *PAL* and *peroxidases* were induced by exogenous NO under LT stress, while *CHS* and *CHI* were down-regulated by both LT stress and NO + LT treatment in cucumber seedlings. Furthermore, *PAL* was regulated by HD-ZIP, and *peroxidase* was regulated by both WRKY21 and MYB63 TFs. This result was consistent with the previous study that suggested that some TFs including MYB, WRKY, and HD-ZIP could control the lignin biosynthesis [53]. In addition, 61 *WRKY* genes were identified according to the cucumber (Chinese Long, 9930) genome (v3.0) [54], and *WRKY46* conferred cold tolerance to transgenic plants and positively regulated the cold signaling pathways in an ABA-dependent manner in cucumber seedlings [16]. We found that WRKY21 was involved in improving the cucumber LT tolerance in the present study. These results suggested that exogenous NO could participate in inducing LT tolerance of cucumber by regulating the expression of WRKY21 and MYB63, and these TFs could further regulate the phenylalanine metabolism and lignin pathways, thereby improving the LT tolerance of cucumber.

NO interacted with phytohormones involved in the regulation of plant growth and development, pathogen defense, and abiotic stress responses in plants [55]. A study showed that NO was required for the transcription of *GA3ox1* and *GA3ox2*, two key biosynthetic enzymes for active GA [56]. We found that exogenous NO could regulate the transcription of *GA3ox* under LT stress in cucumber seedlings. SA and ethylene have also been indicated to interact with NO to regulate plant growth and improve the abiotic stress tolerance in plants. For example, SA can raise the content of NO, and NO interacts with SA to play a crucial role in regulating stomatal closure in *Arabidopsis* leaves and alleviate cadmium toxicity in rice [57,58]; The nature of NO-ethylene crosstalk can be synergistic and also antagonistic, interestingly, most of the growth and developmental processes (e.g., fruit ripening, de-etiolation) are regulated by NO-ethylene antagonism, while in abiotic stress responses the picture is more complex [59]. In this study, we found that LT activated the ethylene and SA response mechanism to stress by up-regulating the expression of *ERF1*, *MKK4_5*, *ETR*, *EIN3* and *NPR1*, *TGA*, *PR1* genes. In addition, exogenous NO can also regulate the content of GA by up-regulating the expression level of *GA3oX*, and similar results were reported previously [56]. Furthermore, the *E**IN3*, *NPR1*, and *GA3oX* were regulated by WRKY21, b-ZIP, and HD-ZIP TFs, respectively. According to these results, we supposed that the WRKY21, b-ZIP, and HD-ZIP were activated by LT stress, and these TFs initiated the mechanisms of the hormone response to stress by inducing the expression of genes related to ethylene, GA, and SA. In addition, exogenous NO could delay the growth of cucumber plants by participating in HD-ZIP-regulated GA metabolism under LT stress, thereby improving the LT tolerance of cucumber. A similar study reported that LT stress led to a decrement in bioactive GAs, thus inhibiting rice seed germination [60].

The ability of plant cells to undergo transformation and regeneration is associated with cell cycle activity [61], while the mini-chromosome maintenance protein [MCM (2-7)] complex is associated with helicase activity for replication fork formation during DNA replication [62]. So the MCM2-7 plays an essential role in the plant cell cycle. A study also reported that the licensing of origins and the loading of MCM2-7 onto DNA is restricted to late mitosis and the G1 phase of the cell cycle [63]. The *MCM5* and *MCM6* were up-regulated by exogenous NO under LT stress in this study. In addition, the *MCM5* and *MCM6* were regulated by HD-ZIP. A previous study reported that *MCM6* shuttled between cytoplasm and nucleolus in a cell cycle-dependent manner, and the subunits of the *MCM2-7* were coordinately expressed during *Arabidopsis* development and were abundant in proliferating and young tissues [64]. Combined with previous research, we speculated that exogenous NO activated down-regulation of HD-ZIP, further negatively regulated the *MCM5* and *MCM6*, and finally promoted cell cycle transition, cell mitosis, and accelerated the production of the new cell to resist the damage of LT to cucumbers in this study.

The light-harvesting chlorophyll a/b binding (LHC) proteins, namely antenna proteins, play a significant role in capturing solar energy, as well as in photoprotection under stress conditions [65]. When plants were exposed to stress conditions that might generate photo-oxidative damage, these proteins assume a conformation able to dissipate the excess energy excitation as heat [66]. We found that LT treatment down-regulated the expression level of *LHCB1*, *LHCB*3, *LHCB5*, and *LHCA1*, and these genes were also negatively regulated by MYB63 under LT stress. In addition, exogenous NO down-regulated the expression levels of *LHCB1*, *LHCB*3, *LHCB5*, and *LHCA1* by negative regulating the b-ZIP. Similar results were also reported by a previous study [67], which found that the levels of the *LHCA1-4* subunits are reduced under Fe deficiency, and it impairs the plant light-harvesting capacity, resulting in decreased photosynthetic efficiency in rice seedlings. Consistent with these results, we propose that LT reduced the photosynthesis by down-regulating the expression levels of *LHCB1*, *LHCB*3, *LHCB5*, and *LHCA1* gene, and NO could not improve the photosynthesis of cucumber seedlings at the transcriptional level.

## 4. Materials and Methods

### 4.1. Plant Materials, Growth Conditions, and LT Treatments

Cucumber seeds of Jinyan No. 4 (purchased from Shandong Xiangyun seed company, China) were germinated at 28 °C for 24 h in dark and then sown in vermiculite at (25 ± 1) °C/(20 ± 1) °C (day/night). After the two cotyledons were fully expanded, healthy seedlings were selected and transplanted into plastic containers (diameter × height, 120 × 110 mm) filled with a mixture of peat and vermiculite (2:1 by volume). Since 7 days after transplantation, 50 mL of Hoagland’s nutrient solution was irrigated every four days. The conditions were maintained as follows: (25 ± 1) °C/(18 ± 1) °C (day/night), approximately 75% relative humidity, 14/10 h (light/dark) photoperiod achieved with supplemental lights, and 300 μmol·m^−2^·s^−1^ average photosynthetic photon flux density (PPFD) across replications for daytime hours. When the second true leaf is fully expanded, that is, after fifteen days of pre-culture, different treatments were carried out. The cucumber seedlings were divided into three groups, two of which were pre-treated with distilled water and the other group was pre-treated with 200 μmol·L^−1^ SNP (Sodium Nitroprusside, as a donor of exogenous NO, purchased from Sigm, USA). After the cucumber seedlings were pre-treated with distilled water and SNP for 2 days. SNP-treated group and one of the distilled water-treated group were placed in an incubator (Percival, Perry, IA, USA) with the temperature of (10 ± 1) °C/(6 ± 1) °C (day/night), approximately 75% relative humidity, 14/10 h (light/dark) photoperiod, and 100 μmol·m^−2^·s^−1^ average PPFD at 10 o’clock in the morning on the third day, and another group of cucumber seedlings treated with distilled water grew in another incubator with normal conditions. The experiment for transcriptome was defined as: (1) Control, cucumber seedlings treated with distilled water and grew in normal conditions for 24 h; (2) LT, cucumber seedlings treated with distilled water and then grew under LT stress for 24 h; (3) NO + LT, cucumber seedlings pre-treated with SNP and then grew under LT stress for 24 h. The experiment was arranged in a completely randomized design with three replicates.

### 4.2. Measurement of Physiological Parameters

The chilling damage index (CI) was measured after low-temperature stress of 24 h according to the method described by [68]. The degrees of low-temperature tolerance were classified into six grades: 0, no symptom; level 1, chlorosis or crinkled at the cotyledon; level 2, chlorosis or crinkled at the edge of old leaves, level 3, chlorosis or crinkled at the edge of functional leaves with good new leaves; level 4, chlorosis or crinkled and wilting of functional leaves with damaged new leaves; level 5, severe damage of new leaves, plants wilt or dead. The symptoms of cucumber seedlings with different treatments were shown in Figure 1. The CI was calculated according to the following formula:$$\mathrm{CI}=\frac{\sum \;\left(\mathrm{each}\;\mathrm{level}\;\times \;\mathrm{number}\;\mathrm{of}\;\mathrm{plants}\;\mathrm{with}\;\mathrm{corresponding}\;\mathrm{level}\right)}{\mathrm{the}\;\mathrm{highest}\;\mathrm{level}\;\times \;\mathrm{total}\;\mathrm{number}\;\mathrm{of}\;\mathrm{treated}\;\mathrm{plants}}$$

The malondialdehyde (MDA) content was determined in terms of thiobarbituric acid reactive substances (TBARS) according to the method described by Heath and Packer [69]. The relative electrolyte leakage was measured according to the method of Xu et al. [70]. The maximum photochemical efficiency (Fv/Fm) was measured using an Imaging-PAM (IMAG-MAX; Walz, Germany) according to our previous study [10].

### 4.3. RNA Extraction, RNA-seq Library Construction, and Sequencing

Total RNA of samples were extracted from frozen leaf tissues of cucumber seedlings and then its purity and integrity were detected using Nanodrop 2000 and 2% agarose gel electrophoresis, respectively. The quality-controlled mRNA samples were used for strand-specific RNA library construction. mRNA was purified from total RNA and enriched by A-T complementary pairing with poly-T oligo-attached magnetic beads. Then the fragmentation buffer was added to break the mRNA into short fragments. mRNAs were used as a template, six-base random hexamers were used to synthesize one-strand cDNA, and then the buffer, dNTPs, and DNA polymerase I were added to synthesize two-strand cDNA. AMPure XP beads were used to purify the double-strand cDNA. The purified double-stranded cDNA is then repaired, A-tailed, and ligated with a sequencing adapter, then AMPure XP beads were used for fragment size selection, and finally, PCR enrichment was performed to obtain the final cDNA library. The quality-controlled libraries were used for transcriptome sequencing. Paired ends of 250–300 bp sequencings were carried out on the HiSeq platform.

### 4.4. Transcriptomic Analysis

The quality of clean reads (clean data) was evaluated using Fastqc (https://www.bioinformatics.babraham.ac.uk/projects/fastqc/, accessed on 15 April 2021), removing poly-N, low-quality, and adaptor sequences [71]. Cucumber (Chinese Long) v3 reference genome, transcript files, and annotation files were downloaded in GuGenDB (http://cucurbitgenomics.org/, accessed on 15 April 2021), and Salmon was used to map the high-quality sequences with cucumber genome and transcripts at the same time to obtain gene expression (TPM) [72]. DESeq2 was used for standardization and differential expression analysis [73]. The |log_2_ (Fold Change)| > 1 and the P_adj_ < 0.01 as the standard to identify the differential expression genes in each comparison. The Gene Ontology (GO) and Kyoto Encyclopedia of Genes and Genomes (KEGG) enrichment analysis was carried out using the clusterProfiler package in R. KEGG annotation information was obtained from the KAAS comparison, and the cluster Profiler package was used for GO enrichment and KEGG pathway analysis [74]. The transcription factors in differentially expressed genes were predicted using PlantRegMep [75]. Furthermore, the EAT-upTF v0.1 was used to establish a gene regulatory network (http://chromatindynamics.snu.ac.kr:8080/EatupTF, accessed on 15 April 2021).

### 4.5. Quantitative Real-Time PCR (qRT-PCR) Analysis

The total RNA was isolated using Trizol reagent according to the manufacturer’s instructions, and DNase I was used to remove any gDNA contamination. The reverse transcriptase (Takara, China) was used to reverse-transcribe pure RNA to cDNA according to the manufacturer’s protocol. The genes involved in the phytohormone signal transduction, cell cycle, and phenylalanine metabolism pathways were selected for qRT-PCR analysis. The primers were generated in the online software Primer 3.0 (http://bioinfo.ut.ee/primer3-0.4.0/, accessed on 4 September 2021) and the pair of primers were presented in Appendix A. *Actin* was used as the reference gene. qRT-PCR was performed using SYBR^®^ Premix Ex Taq™ (Toyobo, Osaka, Japan). The PCR reactions were carried out on the iCycler iQ Multicolor Real-time PCR Detection System (Bio-Rad, Hercules, CA, USA). The relative expression level of each gene was calculated by 2^−ΔΔCt^ with three technical replicates [76]. All experiments had three replicates with three technical replicates.

### 4.6. Statistical Analysis

The statistical software package SPSS Statistics 19.0 (SPSS, Inc., Chicago, IL, USA) was used for statistical analyses. Data was performed by two-tailed analysis of variance (ANOVA), and values represent the means ± SE of at least three replicates. *p* < 0.05 was considered to be differentially significant.

## 5. Conclusions

In the present study, the transcriptome analysis of cucumber seedlings revealed the molecular mechanism of plant response to LT and the mitigation effect of exogenous NO on LT stress in cucumbers. The transcript level of genes related to the cell cycle, photosynthesis, flavonoid accumulation, lignin synthesis, active GA, phenylalanine metabolism, phytohormone ethylene, and SA signaling were significantly regulated by LT stress. Exogenous NO can improve the LT tolerance of cucumber seedlings by affecting Fv/Fm, chilling damage index, electrolyte leakage, lipid peroxidation, and changing the phenylalanine metabolism, lignin synthesis, plant hormone (SA and ethylene) signal transduction, and cell cycle pathway. In addition, four differentially expressed transcription factors MYB63, WRKY21, HD-ZIP, and b-ZIP were identified which can regulate their target genes such as *LHCA1*, *LHCB1*, *LHCB3*, *LHCB5*, *CSH*, *EIN3, peroxidase*, *PAL*, *MCM5*, *MCM6*, *GA3ox*, and *NPRI*, to modulate plant responses to exogenous NO under LT stress. It is worth mentioning that HD-ZIP and b-ZIP specifically responded to exogenous NO under LT stress. These results demonstrate that exogenous NO improved the LT tolerance of cucumber plants by modulating the transcription of some key TFs and their downstream genes, thereby regulating photosynthesis, lignin synthesis, plant hormone signal transduction, phenylalanine metabolism, cell cycle, and GA synthesis. Our study unveiled potential molecular mechanisms of plant response to LT stress and indicated the possibility of NO application in cucumber production under LT stress, particularly in winter and early spring.

## Figures and Tables

**Figure 1 ijms-23-05615-f001:**
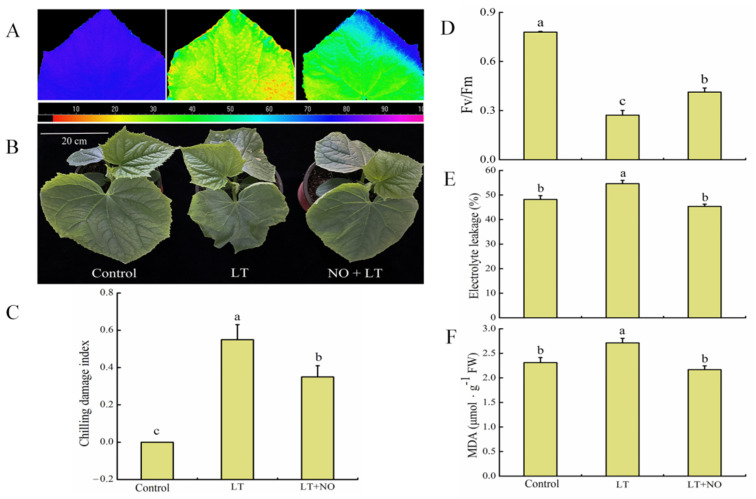
Effects of low temperature (LT) on photosynthetic apparatus and membrane stability in cucumber leaves as influenced by nitric oxide (NO) pretreatment. (**A**) Fv/Fm image, (**B**) plant phenotype, (**C**) chilling damage index, (**D**) Fv/Fm values, (**E**) electrolyte leakage, and (**F**) MDA content in cucumber leaves. Control, the cucumber seedlings were sprayed with distilled water and incubated under normal conditions; LT, the cucumber seedlings were sprayed with distilled water and incubated at 10 ± 1 °C/6 ± 1 °C (day/night) temperatures; NO + LT, the cucumber seedlings were sprayed with 200 μmol·L^−1^ SNP and incubated under LT condition. Data are an average of 6 replicates ± SE. Means denoted by different letters indicate a statistically significant difference at *p* < 0.05.

**Figure 2 ijms-23-05615-f002:**
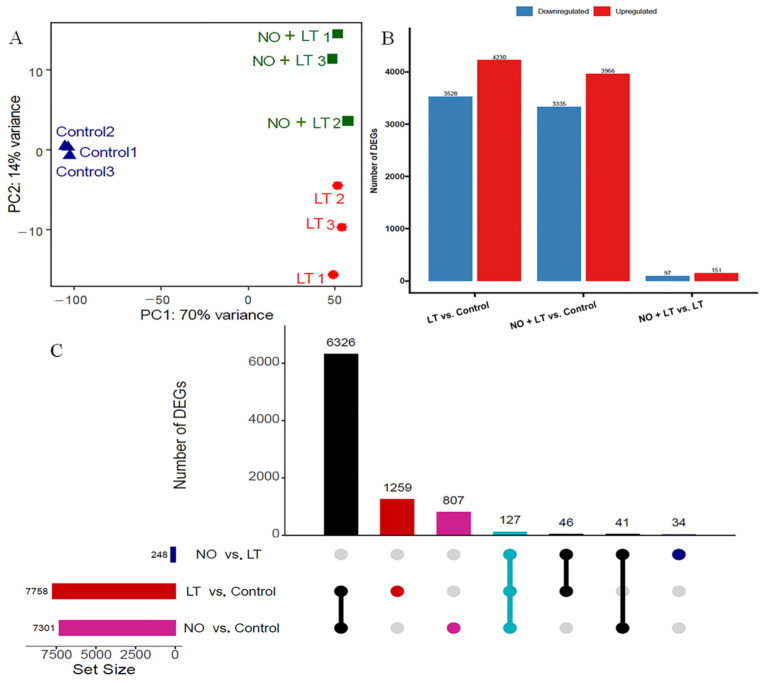
The differentially expressed genes (DEGs) in cucumber seedlings as influenced by nitric oxide (NO) pretreatment under low temperature (LT) stress. (**A**) Principal Component Analysis (PCA) of log_2_-transformed normalized read counts of each RNA-seq dataset of Control, LT, and NO + LT; (**B**) number of DEGs in LT vs. Control, NO + LT vs. Control and NO + LT vs. LT; (**C**) UpSet diagram for the DEGs obtained from the comparison of the three groups, the figures showing the sizes of DEG sets (left) and the combination matrix at the bottom shows the intersections between the sets and the bar above it encodes the size of the intersection.

**Figure 3 ijms-23-05615-f003:**
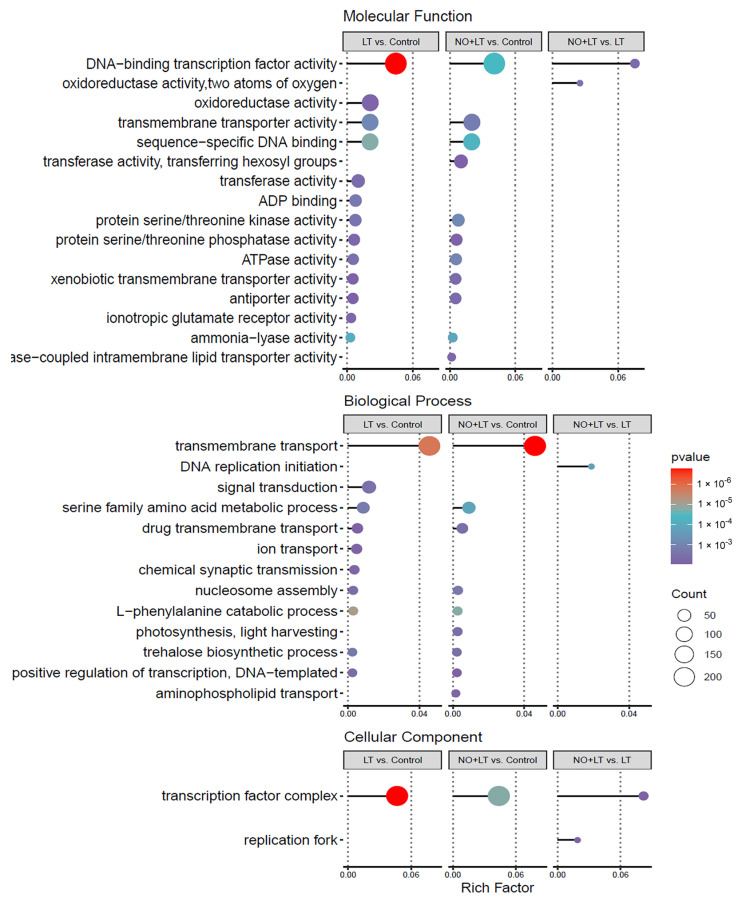
Gene ontology (GO) enrichment analysis of differentially expressed genes (DEGs) for LT vs. Control, NO + LT vs. Control, and NO + LT vs. LT. The color gradient represents the size of the *p*-value and the size of the circle represents the number of DEGs. The BP, CC, and MF represent biological processes, cellular components, and molecular functions, respectively.

**Figure 4 ijms-23-05615-f004:**
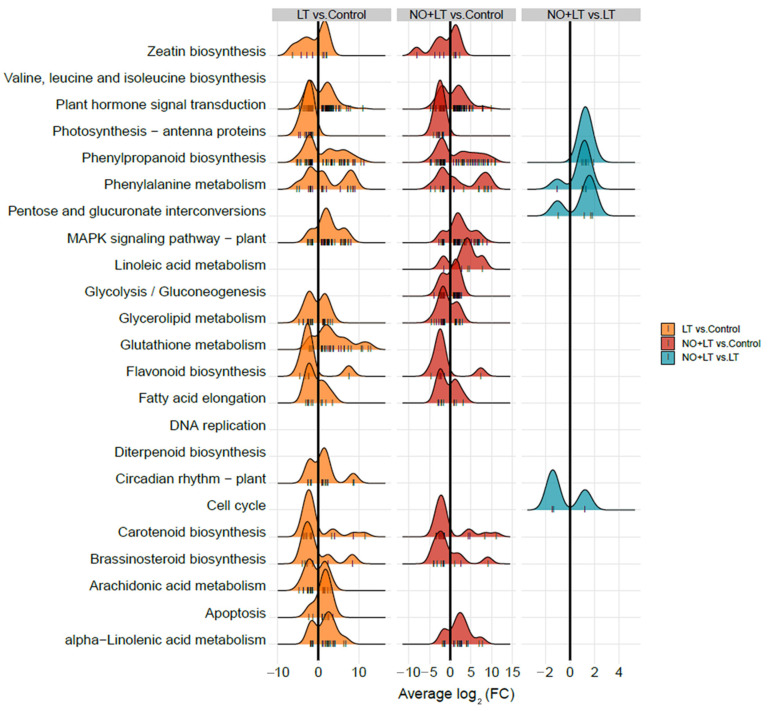
Ridgeline plot of Kyoto Encyclopedia of Genes and Genomes (KEGG) enrichment analysis. The X-axis represents the average of the gene’s log_2_ (fold change). Greater than zero is the up-regulated gene, and less than zero is the down-regulated gene. Ridge height reflects the number of DEGs enriched in this pathway.

**Figure 5 ijms-23-05615-f005:**
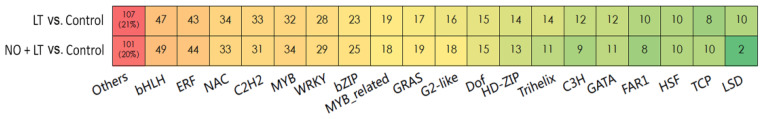
The types and numbers of TF families in the DEGs obtained by LT vs. Control and NO + LT vs. Control, respectively. Color changes represent the changes in gene number.

**Figure 6 ijms-23-05615-f006:**
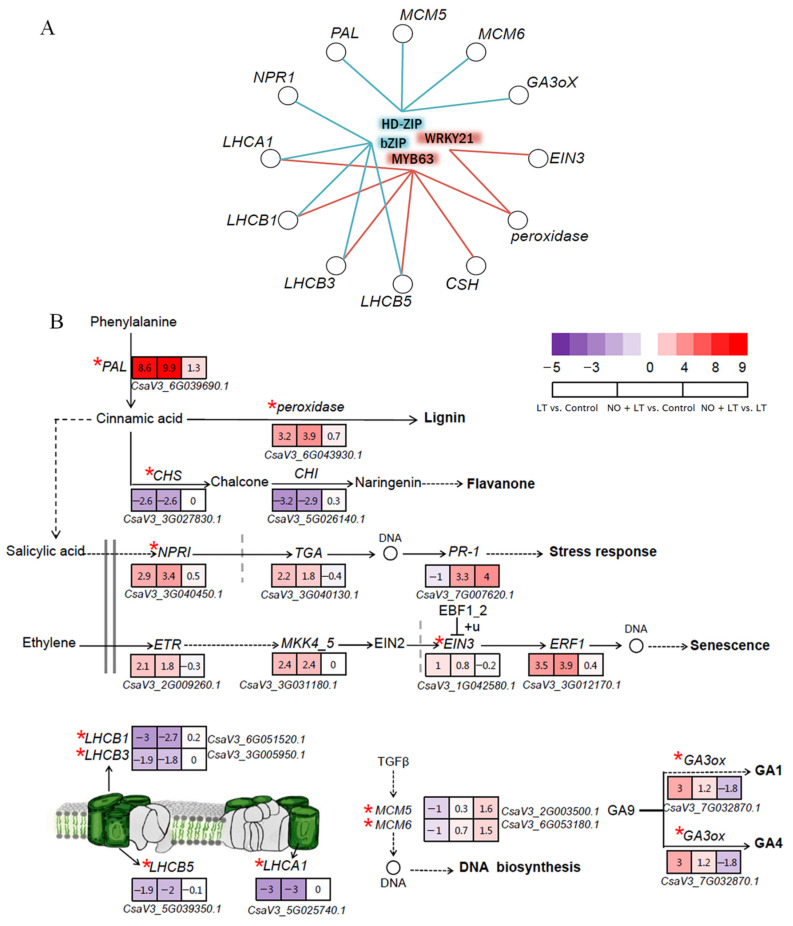
(**A**) A diagram of TFs regulatory network. Blue represents TFs enriched only after NO application under LT stress. Red represents the TFs enriched in both NO and LT. (**B**) Schematic diagram of main pathways and key gene transcript abundance after exogenous application of NO. The Heatmap shows the level of gene expression in different comparisons. The color from red to purple represents the DEGs up-regulated to down-regulated. The gene ID is at the bottom. ‘*’ represents the genes regulated by transcription factors.

**Figure 7 ijms-23-05615-f007:**
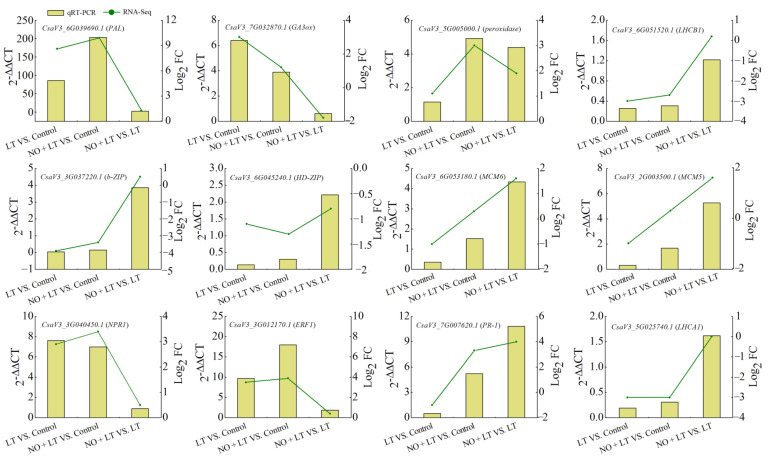
Validation of 12 selected genes by qRT-PCR. The mRNA expression levels were calculated by 2^−^^△△CT^, and the Actin gene was used for normalizing the expression level. The ratio of relative expression in each combination is shown in the figure. The accumulated transcripts of the candidate genes in Log_2_ (Fold Change) were obtained from the RNA-seq data.

**Table 1 ijms-23-05615-t001:** The upstream transcription factors (TFs) enrichment results.

	BBR-BPC	GRAS	AP2	b-ZIP	Dof	BES1	C2H2	M YB	MYB-Related	HD-ZIP
LT vs. Control	1.24 × 10^−41^	7.56 × 10^−38^	6.40 × 10^−32^	5.22 × 10^−16^	1.82 × 10^−15^	5.57 × 10^−14^	5.07 × 10^−12^	9.12 × 10^−5^	1.44 × 10^−5^	-
NO + LT vs. Control	7.08 × 10^−40^	6.57 × 10^−32^	8.88 × 10^−29^	6.13 × 10^−12^	8.84 × 10^−12^	5.45 × 10^−12^	1.96 × 10^−10^	9.96 × 10^−5^	4.12 × 10^−4^	1.04 × 10^−4^

Note: The upstream TFs were obtained by LT and NO + LT compared with the Control. The significance of the difference is represented by the *p*-value.

**Table 2 ijms-23-05615-t002:** The key DETFs in LT vs. Control, NO + LT vs. Control and NO + LT vs. LT.

TF	Tair Gene ID	Cumuber Gene_ID	*p*-Value	Control	LT	NO + LT	LT vs. Control	NO + LT vs. Control	NO + LT vs. LT
MYB63	AT1G79180.1	CsaV3_7G004040.1	0.01075	0.015	0.065	0.167	2.1	3.5	1.4
WRKY21	AT2G30590.1	CsaV3_2G013650.1	0.014038	1.197	5.064	3.067	2.1	1.6	−0.7
b-ZIP	AT2G36270	CsaV3_3G037220.1	0.016527	19.248	1.29	1.790	−3.9	−3.4	0.5
HD-ZIP	AT2G22430.1	CsaV3_6G045240.1	0.033767	192.76	89.45	79.388	−1.1	−1.3	−0.8

Note: Tair gene ID represents the gene ID encoding four TFs in Arabidopsis. The number represents the value of log_2_ (fold change).

## Data Availability

Not applicable.

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
