# Peer review of "Unveiling Molecular Mechanisms of Nitric Oxide-Induced Low-Temperature Tolerance in Cucumber by Transcriptome Profiling"

_ijms, 2022, doi:10.3390/ijms23105615_

Round 1

Reviewer 1 Report

This work is a large-scale study aimed to explore the molecular mechanism of plant response to low temperature stress and the mitigation effect of exogenous NO in cucumber.

This work, in my opinion, can be published in IJMC and will certainly be of interest to readers of the Molecular Plant Sciences section of IJMC.

However, there are minor inaccuracies in the work that should be corrected:

1) On line 45 in the sentence “However, cucumber plants routinely encounter chilling during the winter or early spring production in northern China“ it is probably worth clarifying that this problem is not only in northern China, but also in other countries with a similar climate.

2) In the results section, there is no reference to Figure 1D in the text. Probably it should be put on line 107 or 108 at the discretion of the authors.

3) There are several inaccuracies with references to Figure 2, namely, on line 134 should be fig.2B instead of fig.1B, and it turns out that line 135 does not need a reference to fig.2. In addition, there is no link to fig.2C.

4) As an additional comment, I would like to note that the list of abbreviations is given in the supplementary data. So, it would be convenient if somewhere at the end of the introduction section there was a link to this list of abbreviations.

Author Response

Dear reviewer:

We appreciate the efforts of your careful reviews and insightful comments on our manuscript. We hope that we have addressed all the concerns raised by you. Our response to your comments has been appended below point-by-point.

Comments:

This work is a large-scale study aimed to explore the molecular mechanism of plant response to low temperature stress and the mitigation effect of exogenous NO in cucumber. This work, in my opinion, can be published in IJMC and will certainly be of interest to readers of the Molecular Plant Sciences section of IJMC.

Response: Many thanks for evaluating the merits of our manuscript. Your valuable comments helped us a lot to improve the quality of the manuscript.

Point 1: On line 45 in the sentence “However, cucumber plants routinely encounter chilling during the winter or early spring production in northern China“ it is probably worth clarifying that this problem is not only in northern China, but also in other countries with a similar climate.

Response 1: As suggested, we changed the sentence ‘However, cucumber plants routinely encounter chilling during the winter or early spring production in northern China’ to ‘However, cucumber plants routinely encounter chilling (0-12.5 °C) during the winter or early spring production in regions with cold climates such as in northern China’.

Point 2: In the results section, there is no reference to Figure 1D in the text. Probably it should be put on line 107 or 108 at the discretion of the authors.

Response 2: As suggested, we have cited the ‘Fig 1D’ in lines 107 and 108.

Point 3: There are several inaccuracies with references to Figure 2, namely, on line 134 should be fig. 2B instead of fig.1B, and it turns out that line 135 does not need a reference to fig. 2. In addition, there is no link to fig. 2C.

Response 3: As suggested, we replaced fig. 1B with fig. 2B in line 134, and deleted the fig. 2 in line 135. In addition, we have cited Fig. 2C in line 143.

Point 4: As an additional comment, I would like to note that the list of abbreviations is given in the supplementary data. So, it would be convenient if somewhere at the end of the introduction section there was a link to this list of abbreviations.

Response 4: As suggested, we added a link to the list of abbreviations at the end of the introduction section.

Reviewer 2 Report

The manuscript by Wu and co-authors demonstrates the molecular mechanisms induced by nitric oxide for low-temperature stress tolerance in cucumber. The study has shown molecular mechanisms of plant response to low-temperature stress and how nitric oxide application resulted in low-temperature stress tolerance via the induced molecular mechanisms. The manuscript is well written with exhaustive parameters studied for a sound conclusion. There are a few minor inputs required to further improve the readability of the manuscript.

It is suggested to provide productivity loss data due to low-temperature stress in Introduction.

Include 3-4 lines on the role of plant hormones and other signaling molecules in low-temperature stress tolerance before role of nitric oxide.

 All abbreviations are to be explained on the first mention.

Author Response

Dear reviewer:

We appreciate the efforts of your careful reviews and insightful comments on our manuscript. We hope that we have addressed all the concerns raised by you. Our response to your comments has been appended below point-by-point.

Comments:

The manuscript by Wu and co-authors demonstrates the molecular mechanisms induced by nitric oxide for low-temperature stress tolerance in cucumber. The study has shown molecular mechanisms of plant response to low-temperature stress and how nitric oxide application resulted in low-temperature stress tolerance via the induced molecular mechanisms. The manuscript is well written with exhaustive parameters studied for a sound conclusion. There are a few minor inputs required to further improve the readability of the manuscript.

Response: Many thanks for evaluating the merits of our manuscript. Your valuable comments helped us a lot to improve the quality of the manuscript.

Point 1: It is suggested to provide productivity loss data due to low-temperature stress in Introduction.

Response 1: As suggested, we have provided productivity loss data due to low-temperature stress in the Introduction.

Point 2: Include 3-4 lines on the role of plant hormones and other signaling molecules in low-temperature stress tolerance before role of nitric oxide.

Response 2: As suggested, we revised it.

Point 3: All abbreviations are to be explained on the first mention.

Response 3: As suggested, we defined all abbreviations in the first mention throughout the manuscript.
